# Localization of the Neuropeptide Arginine Vasotocin and Its Receptor in the Osmoregulatory Organs of Black Porgy, *Acanthopagrus schlegelii*: Gills, Kidneys, and Intestines

**DOI:** 10.3390/ijms232113421

**Published:** 2022-11-03

**Authors:** Ganesan Nagarajan, Aruna Adimoolam, Yousef Ahmed Alkhamis, Roshmon Thomas Mathew, Ching-Fong Chang

**Affiliations:** 1Basic Sciences Department, PYD, King Faisal University, Hofuf-420, Al-Asha 31982, Saudi Arabia; 2Center of Excellence for the Oceans, National Taiwan Ocean University, Keelung 20224, Taiwan; 3Department of Aquaculture, National Taiwan Ocean University, Keelung 20224, Taiwan; 4Animal and Fish Production Department, College of Agricultural and Food Sciences, King Faisal University, Hofuf-420, Al-Asha 31982, Saudi Arabia; 5Fish Resources Research Center, King Faisal University, Hofuf-420, Al-Asha 31982, Saudi Arabia

**Keywords:** *avt*, *avtr*, *α-nka*, gill, black porgy

## Abstract

The neurohypophysial hormone arginine vasotocin (*avt*) and its receptor (*avtr*) regulates ions in the osmoregulatory organs of euryhaline black porgy (*Acanthopagrus schlegelii*). The localization of *avt* and *avtr* transcripts in the osmoregulatory organs has yet to be demonstrated. Thus, in the present study, we performed an in situ hybridization analysis to determine the localization of *avt* and *avtr* in the gills, kidneys, and intestines of the black porgy. The *avt* and *avtr* transcripts were identified in the filament and lamellae region of the gills in the black porgy. However, the basal membrane of the filament contained more *avt* and *avtr* transcripts. Fluorescence double tagging analysis revealed that *avt* and *avtr* mRNAs were partially co-localized with α-Nka-ir cells in the gill filament. The proximal tubules, distal tubules, and collecting duct of the kidney all had positive hybridization signals for the *avt* and *avtr* transcripts. Unlike the α-Nka immunoreactive cells, the *avt* and *avtr* transcripts were found on the basolateral surface of the distal convoluted tubule and in the entire cells of the proximal convoluted tubules of the black porgy kidney. In the intestine, the *avt* and *avtr* transcripts were found in the basolateral membrane of the enterocytes. Collectively, this study provides a summary of evidence suggesting that the neuropeptides *avt* and *avtr* with α-Nka-ir cells may have functions in the gills, kidneys, and intestines via ionocytes.

## 1. Introduction

Arginine vasotocin (Avt) is a peptide released by the neurohypophysis of non-mammalian vertebrates, whereas arginine vasopressin (AVP) is its structural and functional equivalent in mammals [1]. The Avt/AVP family binds to specific plasma membrane receptors and have a wide range of physiological effects including behavior, stress, reproduction, vascular control, and osmoregulation. In teleost fish, *avt* receptors (*avtr*) are distributed throughout a broader range of tissues. Thus, the central nervous system as well as organs such as the gills, kidneys, gonads, and gastrointestinal tract have been found to be express the *avtr* isoform V1 (*avtr1*) [2]. Avt appears to alter the fish hydromineral balance by regulating the movement of ion and water transport across many osmoregulatory tissues [3]. The non-mammalian neuropeptides Avt and isotocin (It), as well as the mammalian AVP and oxytocin families, have been linked to blood pressure regulation and antidiuretic effects in osmoregulation [4,5]. Black porgy, *Acanthopagrus schlegelii*, also known as black sea bream, is one of the commercially valuable species caught in Taiwan’s coastal fisheries [6]. Black sea bream are disease resistant and can withstand a wide range of environmental conditions [7,8]. In comparison with mammalian models, teleosts have several new paralogous genes associated with water and ion regulation, potentially allowing teleosts to exist and thrive in a wide range of aquatic environments [9,10,11]. Similar to oxytocin, the major function of AVT is an antidiuretic hormone that influences a variety of behaviors in organisms. Avt function as neurotransmitters and/or neuromodulators in the central nervous system of fish, and both are known to regulate reproductive processes and a variety of social behaviors [12,13].

The sodium potassium ATPase (Nka) is an important player in the reabsorption of Na^+^, water (indirectly), and other solutes in the kidneys [14,15]. One of the key hormones that induce Na reabsorption in the collecting duct is AVP. AVP causes activation of the Gs/adenylyl cyclase system, increased intracellular cAMP concentration, and PKA activation by binding to avt receptors in the basolateral membrane of main cells in the kidney [16]. In mammals, cAMP, the key secondary messenger of AVP, increases Na^+^-K^+^-ATPase activity by increasing the cell surface expression [17]. In fish, gills are multifunctional tissues that carry out a number of vital processes, such as respiration, ion transport, acid-base balancing, and the excretion of nitrogenous waste [18]. The proximal tubule is made up of brush bordered cells with a prominent apical cytoplasmic rim containing few cell organelles as well as ciliated cells, mucous cells, and dark cells [19]. The major cells of connecting tubules have a brush border and numerous mitochondria [19]. Teleost fish are hypo-osmotic to the medium in seawater and consume seawater to compensate for osmotic water loss. Branchial Nka, which is mostly found in chloride cells, removes Na^+^ taken from the digestive tract or obtained by diffusional influx through the gills [20].

Ion regulation is important in the intestines of marine fish. Because of the ionic disequilibrium of marine fish with their surroundings, the osmoregulatory process necessitates high rates of water ingestion (drinking) to compensate for the drying effect of seawater in the gills [18]. The intestine performs crucial functions in nutritional absorption and as a barrier to the external environment, and after the transition of sea water (SW), it also serves the additional function of maintaining active fluid absorption, which is required in seawater [21,22]. Despite the fact that salinity adaptation is a complex process involving a series of physiological responses to environments with varying ion regulatory requirements, euryhaline teleosts can adjust to a wide range of ambient salinities [23]. The primary function of the kidney in marine teleosts is to conserve water, and urine output is minimal. Furthermore, the osmolality of urine is similar to that of bodily fluid, with Mg_2_^+^, SO_4_^2+^, and Ca^2+^ being the major electrolytes rather than Na^+^ and Cl^-^ [24]. The major enzyme Nka is expressed in animal cells to form an electrochemical gradient that drives ion transport in fish osmoregulatory organs such as gills and kidneys [25].

The black porgy, *Acanthopagrus schlegelii*, is a euryhaline teleost that migrates from near-shore shallow areas to coastal waters near land or in estuaries during its transition from the larval to juvenile stage. The current study, the black porgy was used as an animal model to better understand the integrated responses and regulatory roles of neuroendocrine hormones in response to environmental fluctuations. The gill ionocytes are the targets of *avt* and *avtr*, and are hypothesized as being expressed in these ionocytes. Therefore, the purpose of the study is to investigate the localization of *avt* and *avtr* transcripts in the ionocytes of black porgy’s osmoregulatory organs. The mRNA expression of *avt*, *avtr*, and *a-nka* in the gills, kidneys, and intestines of the black porgy was investigated by RT-PCR analysis. Furthermore, in situ hybridization and immunohistochemstry (dual fluorescence) analysis were performed to examine the localization of *avt* and *avtr* mRNA with α-Nka-ir (immuno reactive) cells in the black porgy’s gills, kidneys, and intestines to confirm the co-localizations of neuropeptide hormones and receptors with α-Nka transporters.

## 2. Results

### 2.1. Tissue Distribution of avt, avtr, and a-nka mRNA in the Gills, Kidneys, and Intestines

RT-PCR revealed the predicted single band of *avt*, *avtr*, and α*-nka* transcripts in the gills, kidneys, and intestines of black porgy with varied levels of expression (Figure 1a). The *avt* transcripts were detected in a unique pattern in all osmoregulatory organs such the gills, kidneys, and intestines. The *avtr* and α*-nka* mRNA were found in abundance in the gills and intestines. The mRNA expression of *avtr* and α*-nka* in the kidney, on the other hand, was low (Figure 1a).

### 2.2. Localization of avt and avtr mRNA in the Gills of the Black Porgy

In situ hybridization in the gill was used to examine the expression of the *avt* and *avtr* genes (Figure 1b–g). In the black porgy, the mRNA expression of *avt* (Figure 1c,d) and *avtr* (Figure 1f,g) hybridization signals were detected throughout the gills, such as in the filament and lamellae. However, *avt* and *avtr* transcripts were found in greater abundance in the basal membrane of the filament (Figure 1d,g). There were no discernable signals in the sense probe images of both *avt* and *avtr* (Figure 1b,e).

### 2.3. Localization of avt and avtr mRNA in the Kidneys and Intestines of the Black Porgy

This study found positive hybridization signals for *avt* and *avtr* transcripts in the proximal tubules, distal tubules, and collecting duct of the kidney (Figure 2b–e,g–j). Both *avt* and *avtr* transcripts were localized in the basolateral surface of the distal tubule in the black porgy (Figure 2d,e,i,j). In the proximal tubule, the *avt* and *avtr* transcripts were localized in the whole cell of the tubule (Figure 2b,c,g,h). In the intestine, the *avt* (Figure 3b,c) and its receptors (Figure 3e,f) were found in the basolateral membrane (BLM) of the enterocytes. There were no discernable signals in the sense probe images of both *avt* (Figure 2a and Figure 3a) and *avtr* (Figure 2f and Figure 3d).

### 2.4. Immunolocalization of a-Nka-ir Cells in the Gills, Kidneys, and Intestines of the Black Porgy

The immunohistochemistry was performed to localize the α-Nka-ir cells in the gills (Figure 1h), kidneys (Figure 2k,l), and intestines (Figure 3g,h) of the black porgy. The α-Nka-ir cells were found in the gill filament, particularly in the mitochondria rich cells (Figure 1h). In the kidney, the α-Nka-ir cells were found in the proximal convoluted tubule’s basolateral surface (Figure 2k). Those α-NKa-ir, on the other hand, were found in the entire cells of the black porgy kidney’s distal convoluted tubules (Figure 2l). The α-Nka-ir cells were found in the BLM and BBM of the enterocytes in the black porgy’s intestine (Figure 3g,h).

### 2.5. Co-Localization of avt and avtr mRNA with a-Nka-ir Cells in the Gills, Kidneys, and Intestines of the Black Porgy

According to the fluorescence double tagging analysis, the *avt* (Figure 4c) and *avtr* mRNA (Figure 4f) with α-Nka-ir cells (Figure 4b,e) were partially co-localized (Figure 4a,d) in the black porgy’s gill filament. The *avt* (Figure 5a,c,e) and *avtr* (Figure 5h,j,l) transcripts were identified in the distal convoluted tubule’s basolateral surface comared with the apical membrane, unlike the α-Nka-ir cells (Figure 5b,g,i,n). The *avt* (Figure 5a,c,d) and *avtr* (Figure 5h,j,k) transcripts, on the other hand, were detected in the whole cells of the proximal convoluted tubules in the kidney (Figure 2b,c,g,h). However, the *avt* (Figure 5a,d,e,f,g) and *avtr* (Figure 5h,k,l,m,n) transcripts were partially co-localized with α-Nka-ir cells (Figure 5b,i) in the proximal and distal convoluted tubules (Figure 5a,h). The *avt* (Figure 6a,c,d,f) and *avtr* (Figure 6g,i,j,l) hybridization signals were detected in the basolateral membrane of the enterocytes of the intestine. These transcripts were well co-localized with α-Nka-ir cells (Figure 6b,e,h,k) in the intestine BLM of the enterocyte cells (Figure 6a,c,d,f,g,i,j,l).

## 3. Discussion

This study examined the distribution of *avt* and *avtr* mRNA in the gills, kidneys, and intestinal tissues of the black porgy. The gills and intestines, but not the kidneys, were shown to have high expression levels of the *avtr* and *α-nka* transcripts. The *avt* mRNA, on the other hand, was consistently expressed in the gills, kidneys, and intestines. The *avt* and *avtr* mRNA hybridization signals were identified in the apical membrane of the branchial epithelial cells, the proximal and distal tubule of the kidneys, and enterocytes of the intestines in the euryhaline black porgy. Immunohistochemistry was performed to localize α-Nka immunoreactivity in the chloride cells of the gills, basolateral surface of the proximal tubule and whole cells of the distal tubules in the kidneys, and basolateral membrane of the enterocytes in the intestines of the black porgy. Furthermore, fluorescence dual staining revealed that the *avt* and *avtr* hybridization signals were co-localized with α-Nka-ir cells in the gills, kidneys, and intestines of the black porgy.

The neuropeptide *avt* has been shown to regulate various molecular components of chloride cells, including the Na^+^/K^+^-ATPase pump, the cystic fibrosis transmembrane conductance regulator, and other ionocytes in the gilthead seabream, *Sparus aurata* [26]. The *avt* also regulates ion exchange across the gill epithelium, which is the primary site of ion uptake in the freshwater and secretion in the seawater [3]. The *avt* neuroendocrine system regulates many aspects of fish physiology and behavior, including circadian and seasonal biology, stress responses [27], metabolism, reproduction [28], cardiovascular function, and osmoregulation [29]. As previously mentioned, avt can significantly alter blood flow through the gills, which indirectly affects ion fluxes across the epithelia [30,31]. However, it has been proposed that *avt* has a direct effect on gill ion and water transport [4,32]. Transcripts for *avt* and *avtr* were found in the filament and lamellae regions of the black porgy gill. However, the basal membrane of the filament contained more *avt* and *avtr* transcripts. Interestingly, the *avt* and *avtr* hybridization signals were also detected in mitochondria rich cells. In teleosts, the mitochondria-rich gill cells play a key function in controlling plasma ions [20].

Immunocytochemical studies have shown that the Nka-ir cell is primarily found in the mitochondria-rich gill epithelia of euryhaline teleosts [33]. Nka-ir cells, on the other hand, were found in the epithelia of both the filaments and lamellae in the fresh water (FW)-adapted euryhaline fish [34]. The presence of lamellar Nka-ir cells in some freshwater-adapted euryhaline teleosts is suggested in order to meet the physiological requirement of ion uptake [35,36,37,38,39]. Moreover, based on numerous studies with conventional model species (e.g., salmon, trout, eel, tilapia, and killifish), the ion transport mechanisms in the gill MR cells have been proposed [18,23,40]. Black porgy α-Nka-ir cells were found on the surface of filaments and the interlamellar area. Furthermore, according to our fluorescence double labelling analysis, the *avt* and *avtr* mRNAs were partially co-localized with α-Nka-ir cells in the gill filaments of the black porgy.

In the black porgy kidney, the NKa was immunohistochemically stained using the α-Nka antibody. The immunoreaction was identified in the collecting tubules, distal convoluted tubules, and proximal convoluted tubules. The Nka antibody was found on the basolateral surface of the proximal convoluted tubules, whereas it stained whole cells in the distal convoluted tubules. Similarly, Nka immunostaining was found in the cytoplasm of epithelial cells of distal and collecting tubules, as well as the basolateral of proximal tubular cells of the spotted green pufferfish [24,41]. In the current investigation, the proximal, distal, and collecting ducts of the kidney all showed positive hybridization signals for the *avt* and *avtr* transcripts. Unlike the α-Nka-ir cells, the *avt* and *avtr* transcripts were found in the distal convoluted tubule of the basolateral surface in the kidney. Those transcripts, on the other hand, were found in the entire cells of the black porgy kidney’s proximal convoluted tubules. The proximal tubule reabsorbs 60% of filtered NaCl and water, 90% of filtered bicarbonate, and most essential nutrients such as glucose and amino acids. To facilitate reabsorption, the proximal tubule uses both cellular and paracellular transport systems [42]. The proximal tubule absorbs 65% of the filtered water, as well as Na^+^, Cl, and K^+^. Na^+^ reabsorption accounts for most of the energy consumed by the proximal tubule. Passive apical input through multiple sodium transport routes is enabled by Nkase-energized basolateral active sodium extrusion [16]. Na^+^ enters the cell by enhanced diffusion on the apical surface and is blocked by amiloride.

Intracellular Na^+^ buildup is prevented by the Nkase on the basolateral surface [43]. The enlarged surface area of the apical membrane of the proximal tubular cells supports reabsorptive activities. The brush border, a dense array of microvilli, provides this large surface area, and when combined with leaky tight junctions, high-capacity fluid reabsorption occurs. The distal tubule is a segment with a dense epithelium and low water permeability. It reabsorbs around 5% of the NaCl filtered out of the initial urine. It is also responsible for the regulation of the potassium and calcium filtrate levels, as well as the pH of the filtrate [42]. The AVT and AVP both regulate renal water conservation, although their activities in fish (V1) and mammals (V2) involve in different receptors. Balment et al. [29] investigated the pattern of the AVT V1 receptor expression in the flounder kidney shed light on the potential mechanisms underlying the purported antidiuretic effects. A combination of altered AVT production and target-tissue receptor expression results in osmoregulation [29]. Therefore, the roles of AVT may integrate renal and gill contributions to regulating body fluid homeostasis.

The teleost intestines are essential for nutrition, ion, and water absorption. They play an important role in osmoregulation by compensating for osmotic water loss in a hyperosmotic environment through water reabsorption [25,44]. The tiny intestine epithelium folds internally to produce intestinal crypts, and the epithelium protrudes outside to form the intestinal villi, which are immediately exposed to the luminal environment [45]. The most common type of intestinal epithelial cells (>80%) is enterocytes [46]. Tight connections connect these polarized cells along the intestinal villi, forming a physiological barrier between them. Microvilli, which contain digestive enzymes and nutrient transporters and form the functional intestinal brush border, greatly enhance the surface area of the enterocyte apical membrane [47]. NKA is important for maintaining intracellular homeostasis as it increases the movement of ions and water through a variety of transporters and channels [48]. As Nkase plays a critical role in the absorption of nutrients, ions, and/or water in all settings, it was found in the basolateral membrane of both FW- and SW-acclimated teleosts intestines [49,50,51]. The electrogenic potential is generated by a basolateral α-Nkase, which aids in apical Cl absorption [52]. In agreement with the previous findings, the immunological staining for α-Nka was seen in the enterocytes basolateral membrane of the intestine. In the black porgy intestine, the *avt* and *avtr* transcripts were also found in the basolateral membrane of the enterocytes. The presence of NKa throughout the entire basolateral membrane of enterocytes in the intestine of FW fish [51,53] and SW fish indicates that these cells are involved in food uptake in all habitats [54,55,56,57]. In addition, both the basal and lateral stains were stronger in the enterocytes of SW fish such as marble goby ([51], Atlantic salmon ([57], and rainbow trout [55]. During salinity stress, the corticosteroid receptors were also found in the apical membrane of the branchial epithelial cells of the gills; proximal, collecting, and distal tubules of the kidneys; and enterocytes of the intestine in tilapia to cope with the homeostasis in tilapia [58], suggests that the apical membrane of the epithelial cells, proximal and distal tubules of the kidney, and enterocytes of the intestines all play a role in the osmoregulatory and homeostatic actions of tilapia.

The physiological regulation of body fluid volume and composition is tied closely to *avt*/*avp* throughout the vertebrate series [59]. Our current findings suggest that the neurohypophysial hormone and its receptor may influence ionocyte functions by regulating ionocyte expression based on the co-localization of *avt* and *avtr* with Nka-ir cells. The localization of *avt* and *avtr* transcripts in a variety of osmoregulatory tissues, including the apical membrane of the epithelial cells in the gills, proximal and distal tubules of the kidneys, and enterocytes of the columnar cells in the intestine indicated that the neurohypophysial hormone and receptors play a role in hyper-/hypo-osmotic situations in the black porgy. The co-localization of *avt* and *avtr* with α-Nka-ir cells raised the possibility that neuropeptides function in osmoregulatory organs through ionocytes.

## 4. Materials and Methods

### 4.1. Experimental Fish

Black porgy (6 to 7 months old, *n* = 8) (body weight = 17.69 ± 0.59 g, body length = 9.9 ± 0.37 cm) were obtained from the aquaculture farm and allowed to acclimate in cultured seawater (33 ppt) with a natural light system at the National Taiwan Ocean University aquarium (water temperature, pH, and DO values ranged from 19 to 24 °C, pH 7.5–8.2, and DO 7.1–7.7 mg/L, respectively). The fish tanks were aerated to ensure that the fish had enough dissolved oxygen. Air stones were used to keep the water aerated at all times. In all of the experiments, the level of aeration was sufficient to maintain saturated dissolved oxygen levels. The contents from the fish excrement and uneaten meals were filtered using sponge filters and gravel. The fish were given ad labitum pelleted dry feed at a daily proportion of 1% of their estimated body weight. The fish were beheaded and anesthetized with 1% glycophenol monophenyl ether, and the gill, kidney, and intestine samples were taken and frozen in liquid nitrogen at −80 °C. The experiments followed the principles and procedures of the National Taiwan Ocean University’s Institutional Animal Care and Use Committee (no. 89-91: 98038).

### 4.2. RNA Extraction and Cloning

RNA was extracted from the gills, kidneys, and intestines using the TRIzol^®^ reagent and was reverse transcribed according to the manufacturer’s procedure (Gibco BRL; Grand Island, NY, USA). The resultant cDNA was employed as a template for the PCR amplification of the genes used in the current study. From the cDNA of the black porgy gills, the genes implicated in the stress response were cloned. Multiple alignments of previously known sequences of the corresponding genes were built using the CLUSTAL X program (version 1.81) to identify the conserved area, and the primers were then designed based on this information (Table 1). In a final volume of 25 μL, PCR reactions were done with 2.5 μL of 10X reaction buffer (200 mM Tris-HCl (pH 8.4, 500 mM KCl), 1 μL of 10 mM dNTP, 1 μL of 2 mM MgCl_2_, 0.5 μL each of 10M sense and antisense primers, 1 μL cDNA, and 0.2 μL superscript enzyme (Invitrogen; Cals bad, CA, USA). The following PCR conditions were used: 94 °C for 5 min, 94 °C for 30 s, 50 °C for 30 s, 72 °C for 30 s for 35 cycles, and 72 °C for 10 min. The PCR results were confirmed by electrophoresis on a 1.5% agarose gel stained with ethidium bromide, and DNA fragments were extracted with a Gel-M ^TM^ Gel Extraction system Kit (Bio 101) (VIOGENE; La Jolla, CA, USA) and cloned into the pGEM^®^—T Easy vector (Promega; Madison, WI, USA). A dye terminator cycle sequencing kit (Perkin Elmer; Foster City, CA, USA) was used to sequence the plasmid containing the insert, which was then submitted to BLAST for comparison to known sequences in the NCBI database.

### 4.3. Tissue Distribution of avt, avtr and α-nka in the Gills, Kidneys, and Intestines

The gills, kidneys, and intestines of the black porgy (*n* = 3) was used to extract total RNA. cDNA was synthesized using 1μg of total RNA, oligo (dT)12–18 primers, SuperscriptII, and gene specific primers forthe RT-PCR analysis (Table 1). For the internal control, the *glyceraldehydes 3-phosphate dehydrogenase* (*gapdh*) gene was used. The *gapdh* transcripts did not differ appreciably across the treatments. The PCR settings were 94 °C for 1 min, 55 °C for 1 min, and 72 °C for 2 min for 30 cycles, with the PCR products separated on a 1.5% agarose gel. Reverse transcriptase-free reactions served as the negative controls.

### 4.4. In Situ Hybridization

The *avt* and *avtr* transcripts were localized in the gills, kidneys, and intestines of black porgy using in situ hybridization. The tissues were embedded in paraffin after being fixed in a 4% paraformaldehyde in phosphate-buffered saline (PBS) solution at 4 °C overnight. The paraffin sections (5 μm) were collected (3-aminopropyltriethoxysilane, Sigma, St. Louis, MO, USA) on TESPA-treated slides. The DIG-labelled sense and anti-sense RNA riboprobes of *avt* and *avtr* were made in the black porgy using T7 and Sp6 promoter polymerase (Promega, Madison, WI, USA) on linear plasmid DNA containing corresponding inserts of the genes in the pGEM-T Easy vector, as described previously [60]. The PCR products were purified using a kit (Viogene, Bio 101, La Jolla, CA, USA) and the quantity of the PCR amplification product was determined using spectrophotometry at 260 nm. According to our prior research, the serial sections were rehydrated, prehybridized, and hybridized [61,62]. The sections were then treated overnight at 4 °C with alkaline phosphatase-conjugated sheep anti-digoxigenin antibody (Roche, Penzberg, Germany) (dilution 1:2000 in 2% blocking reagent). NTMT (100 mM NaCl, 100 mM Tris-HCl, pH 9.5, 50 mM MgCl2, 0.1% tween-20) and NBT/BCIP staining were used to visualize the hybridization signals.

### 4.5. Immunohistochemistry

The immunoreactive expression of α-Nka was studied in the gills, kidneys, and intestines of the black porgy using immunohistochemical analysis. In PBS, a few deparaffinized serial slices (5 μm) were treated with 3% H_2_O_2_. The sections were then incubated for 30 min with 1.5% normal goat serum overnight at 4 °C with a commercial primary antibody (α-Nka-IgG) (Santa Cruz Biotechnology Inc., Santa Cruz, CA, USA), followed by 1 h at room temperature with respective anti-rabbit IgG (Vector Laboratories Inc., Burlingame, CA, USA). An ABC kit (avidin-biotin, Vector Laboratories Inc.) and DAB (3, 3**′**-Diaminobenzidine, Sigma) were used to visualize the sections.

### 4.6. Co-Localization of avt and avtr mRNA with α-Nka in the Gills, Kidneys, and Intestines

For the co-localization analysis, first the fluorescence in situ hybridization was done using *avt* and *avtr* specific probes in the gills, kidneys, and intestines of the black porgy. Following hybridization with the respective probes, the examined sections were incubated for 2–3 h with the HNPP (2-hydroxy-3-naphtoic acid-2**′**-phenylanilide phosphate)/FastRED detection kit (Roche) according to the manufacturer’s instructions, and then incubated with an alkaline phosphatase-conjugated sheep anti-digoxigenin antibody (Roche). The sections were washed with running tap water for 5 min to stop the reaction. Second, following in situ hybridization, the immunohistochemistry was performed with the α-Nka antibody in the *avt* and *avtr* hybridized sections of the gills, kidneys, and intestine. The co-localization procedure was done according to our previous research paper [58,63,64]. The hybridized sections (5 μm) were incubated with 3% H_2_O_2_ in PBS for 10 min. After PBS washing, the sections were incubated with 1.5% normal goat serum for 30 min and kept overnight at 4 °C with a commercial primary antibody against α-NKA (α-NKA-rabbit IgG; Santa Cruz Biotechnology Inc., Santa Cruz, CA, USA). After being left overnight, the sections were incubated with fluorophore-conjugated secondary antibody Alexa Fluor 488 (goat anti-rabbit IgG, Invitrogen). The slides were then stained with DAPI as a counterstain (Vector Laboratories, Burlingame, CA, USA). Finally, the Prolong Gold antifade reagent was used to mount the slides (Invitrogen, Molecular Probes, Eugene, OR, USA).

## Figures and Tables

**Figure 1 ijms-23-13421-f001:**
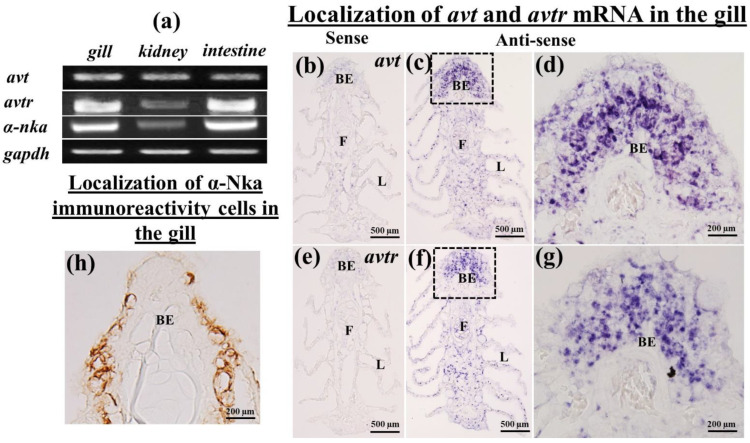
(**a**) The black porgy *avt*, *avtr,* and α-*nka* cDNA consisted of 320 bp, 450 bp, and 968 bp. A single band of the expected size was obtained for *avt, avtr,* and *α-nka* mRNA in the tissues analyzed by RT-PCR. The *gapdh* was used as an internal control. The *avt* mRNA was found to be expressed specifically in the gills, kidneys, and intestines. The *avtr* and α-*nka* transcripts, on the other hand, were strongly expressed in the gills and intestines, whereas in the kidney, the expression was low compared with other osmoregulatory organs. (**b**–**g**) The in situ hybridization was performed to localize the *avt* and *avtr* mRNA in the black porgy’s gills. The hybridization signal for *avt* and *avtr* was detected in both the filaments and lamellae. However, *avt* transcripts were more abundant in the apical membrane of the branchial epithelium compared with the lamellae. There was no hybridization signal detected in the *avt* and *avtr* sense pictures (**e**). (**h**) The immunohistochemistry was performed to localize the α-Nka-ir cells in the gills. The α-Nka-ir cells were found in the filamental mitochondria rich cells. BE: branchial epithelium; F: filament; L: lamellae.

**Figure 2 ijms-23-13421-f002:**
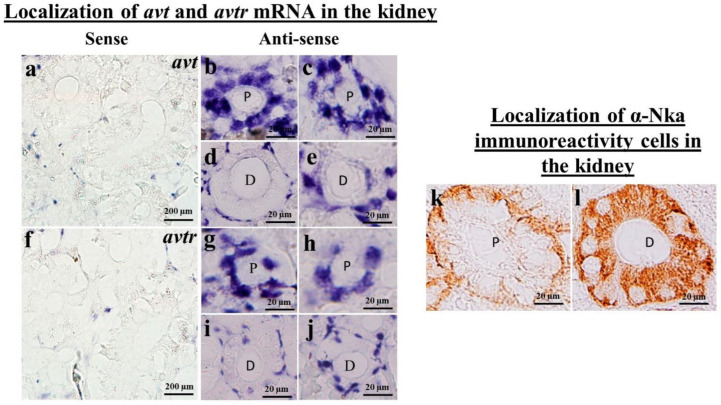
(**a**–**j**) The in situ hybridization was performed to localize the *avt* (**a**–**e**) and *avtr* (**f**–**j**) mRNA in the black porgy’s kidneys. The *avt* and *avtr* hybridization signals were detected in the black porgy kidney’s proximal convoluted (**b**,**c**,**g**,**h**) and distal convoluted tubules (**d**,**e**,**i**,**j**). The *avt* (**d**,**e**) and *avtr* (**i**,**j**) transcripts were found in the distal convoluted tubule’s basolateral surface. Those transcripts, on the other hand, were found in the entire cells of the black porgy kidney’s proximal convoluted tubules (**b**,**c**,**g**,**h**). There was no hybridization signal detected in the *avt* (**a**) and *avtr* sense pictures (**f**). (**k**,**l**) The immunohistochemistry analysis was performed to localize the α-Nka-ir cells in the kidney. The α-Nka-ir cells were found in the proximal convoluted tubule’s basolateral surface. Those transcripts, on the other hand, were found in the entire cells of the black porgy kidney’s distal convoluted tubules. P: proximal convoluted tubules; D: distal convoluted tubules.

**Figure 3 ijms-23-13421-f003:**
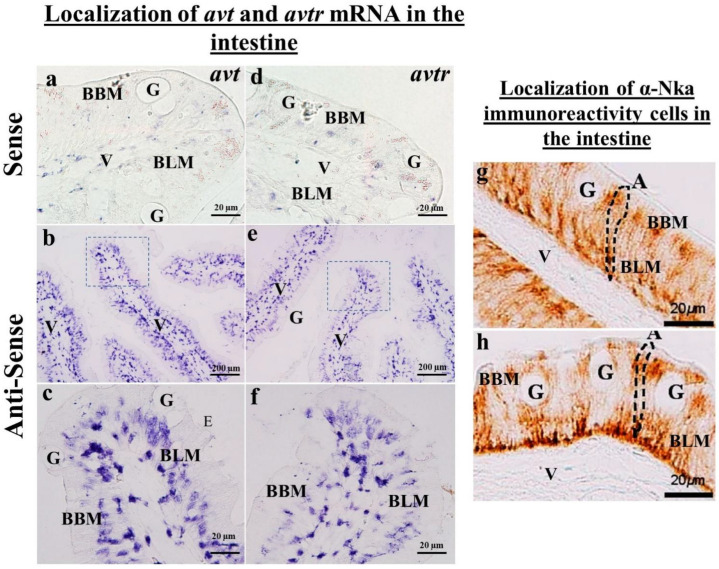
(**a**–**f**) The in situ hybridization was performed to localize the *avt* (**a**–**c**) and *avtr* (**d**–**f**) mRNA in the black porgy’s intestine. The *avt* (**b**,**c**) and *avtr* (**e**,**f**) mRNA expressions were found in the enterocytes BLM of the black porgy intestinal villi. There was no hybridization signal detected in the *avt* (**a**) and *avtr* sense pictures (**d**). The immunohistochemistry was performed to localize the α-Nka-ir cells in the BLW and BBM enterocytes of the black porgy’s intestinal villi (**g**,**h**). BBM: brush border membrane; BLM: basolateral membrane; V: villi; G: goblet cells; A: absotrptive cells.

**Figure 4 ijms-23-13421-f004:**
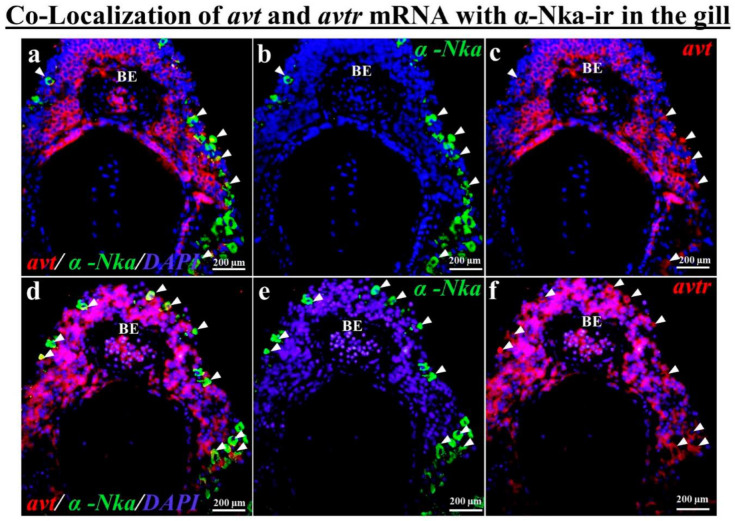
Dual staining (in situ hybridization and immunohistochemistry) was performed to investigate the co-localization of *avt* and *avtr* mRNAs with α-Nka-ir cells together DAPI nuclear staining in the gill of the black porgy (**a**–**f**). The *avt* (**c**) and *avtr* (**f**) transcripts were partially co-localized (**a**,**d**) with the α-Nka-ir cells (**b**,**e**) in the gill filament (apical membrane of the gill branchial epithelium cells). Arrows indicate the co-localized cells of the *avt* and *avtr* transcripts with α-Nka-ir cells. BE: branchial epithelium; F: filament; L: lamellae.

**Figure 5 ijms-23-13421-f005:**
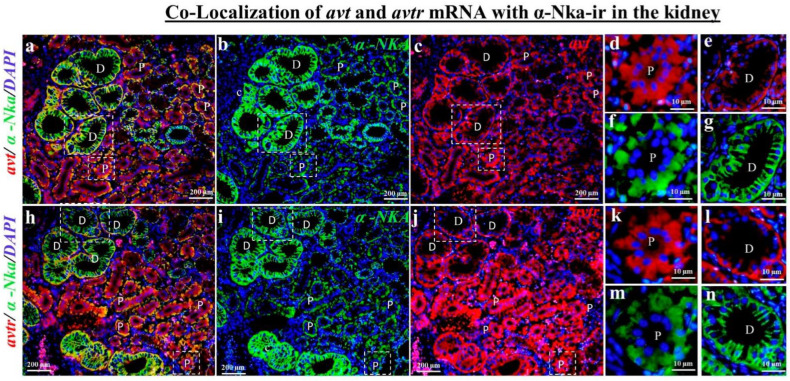
Dual staining (in situ hybridization and immunohistochemistry) was performed to investigate the co-localization of *avt* and *avtr* mRNAs with α-Nka-ir cells together with DAPI nuclear staining in the kidney of the black porgy (**a**–**n**). The *avt* (**c**–**e**) and *avtr* (**j**–**l**) hybridization signals were partially co-localized in the basolateral surface of the both proximal and distal convoluted tubules (**a**,**h**) with the α-Nka-ir cells (**b**,**i**). P: proximal convoluted tubules; D: distal convoluted tubules.

**Figure 6 ijms-23-13421-f006:**
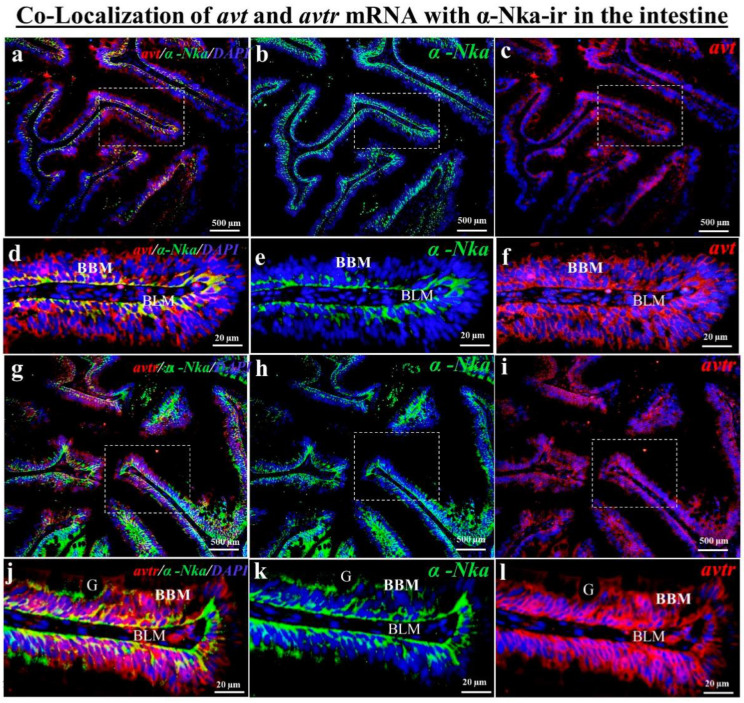
Dual staining (in situ hybridization and immunohistochemistry) was performed to investigate the co-localization of *avt* and *avtr* mRNA with α-Nka-ir cells together DAPI nuclear staining in the intestine of the black porgy (**a**–**l**). The *avt* (**c**,**f**) and *avtr* (**i**,**l**) hybridization signals were co-localized with the α-Nka-ir cells (**b**,**e**,**h**,**k**) in the BLM of the enterocytes of the black porgy’s intestine (**a**,**d**,**g**,**j**). BBM: brush border membrane; BLM: basolateral membrane.

**Table 1 ijms-23-13421-t001:** Oligonucleotide primers used for specific primers for the RT-PCR analysis and in situ hybridization. F: forward, R: reverse, S: sense strand, AS: antisense strand.

Gene	Orientation	Nucleotide Sequence (5′–3′)	Usage
*avt*	F	5′-CTGTGCGTCCTGGGACTCATC-3′	RT-PCR
	R	5′-GGAGCTCTGAGGCATGTTTTG-3′	RT-PCR
	S	5′- CTCCTCCGCCTGCTACATC-3′	in situ
	AS	5′- GGCTGACTCTTCAATCTTC-3′	in situ
*avt-r*	F	5′-CCTGCTGGCAATGTACAACAC-3′	RT-PCR
	R	5′-GGCGCCCAGCAGATAATGTAC-3′	RT-PCR
	S	5′-CAGGTCTCTGGTCTTCAG-3′	in situ
	AS	5′-GCTCTGGATATCGTTGTAAC-3′	in situ
*gapdh*	F	5′-GGCCCCCCTGGCCAAAGT-3′	RT-PCR
	R	5′-TGGGTGTCACCGATGAAG-3′	RT-PCR

## Data Availability

Data are contained within this article. Raw data are available upon request from the corresponding authors.

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
