# Peer review of "Localization of the Neuropeptide Arginine Vasotocin and Its Receptor in the Osmoregulatory Organs of Black Porgy, Acanthopagrus schlegelii: Gills, Kidneys, and Intestines"

_ijms, 2022, doi:10.3390/ijms232113421_

Round 1

Reviewer 1 Report

The authors studied the localization of avt and avtr mRNA in the black porgy gill, kidney, and intestinal tissues. 

The manuscript is interestingly written and includes well-executed documentation.

The article needs a few corrections.

in detail:

In the introduction, authors should specify the purpose of the work.

A moderate revision of the English language is needed

Author Response

Response to Reviewers

We thank the journal editor and two reviewers for their insightful and detailed remarks on our submission. Your recommendations have been taken into account when editing the entire manuscript. A native English speaker has completely reedited the entire manuscript. We revised the reference section according to the journal format. We responded to each reviewer comments and suggestions in the order listed below.

Reviewer # 1

The authors studied the localization of avt and avtr mRNA in the black porgy gill, kidney, and intestinal tissues. 

The manuscript is interestingly written and includes well-executed documentation.

The article needs a few corrections.

in detail:

In the introduction, authors should specify the purpose of the work.

A moderate revision of the English language is needed

Response: Thank you for your kind suggestions. The purpose of the work has been included in the last paragraph of the introduction. A native English speaker has completely reedited the entire manuscript.

Reviewer 2 Report

In the manuscript, the authors have investigated the localization of a neuropeptide avt (arginine vasotocin) and its receptor avtr (arginine vasotocin receptor) mRNA in the gill, kidney, and intestinal tissues of black porgy. The authors have nicely shown that the avtr(s) are highly expressed in the gill and intestine, while it is weekly expressed in the kidney. However, avt mRNA was adequately present in the gill, intestine, and kidney of black porgy. The present study could help to understand the role of the neurophysiological hormone in regulating osmoregulatory organs. In the presented manuscript, the scientific work is interesting and nicely explained. The authors have sufficiently screened the previously published pieces of literature in relevance to the work. The study is well organized and sufficiently discussed. The presented manuscript has enough potential to be considered for publication in my opinion; however, there are some minor issues:

1. In line 77, an abbreviation is used ‘SW’, authors are suggested to introduce its full form before its direct use in the text. This could help to understand the manuscript by non-expert readers of the journal. Please carefully revise the manuscript.

2. In line 110 (Figure 1 legend), the term ‘gadph’ should be replaced with the term ‘gapdh’.

3.  In line 142, it is written – ‘The α-Nka-ir cells There was no hybridization signal was detected in the avt (a) 142 and avtr sense pictures…’. Please revise the sentence.

Author Response

Response to Reviewers

We thank the journal editor and two reviewers for their insightful and detailed remarks on our submission. Your recommendations have been taken into account when editing the entire manuscript. A native English speaker has completely reedited the entire manuscript. We revised the reference section according to the journal format. We responded to each reviewer comments and suggestions in the order listed below.

Reviewer #2

  1. In line 77, an abbreviation is used ‘SW’, authors are suggested to introduce its full form before its direct use in the text. This could help to understand the manuscript by non-expert readers of the journal. Please carefully revise the manuscript.

Response: Thank you very much for your detailed comments on our manuscript. The abbreviations for SW and FW are included in the introduction part.

  1. In line 110 (Figure 1 legend), the term ‘gadph’ should be replaced with the term ‘gapdh’.

Response: We corrected the term “gapdh” in the corrected manuscript.

  1. In line 142, it is written – ‘The α-Nka-ir cells There was no hybridization signal was detected in the avt (a) 142 and avtr sense pictures…’. Please revise the sentence.

Response: We corrected the sentence in the manuscript.